# Application of Fabric Phase Sorptive Extraction (FPSE) Engaged to Tandem LC-MS/MS for Quantification of Brorphine in Oral Fluid

**Dimitra Florou [1], Thalia Vlachou [1,2], Vasilios Sakkas [2] and Vassiliki Boumba [1,*]**

[1] Department of Forensic Medicine and Toxicology, Faculty of Medicine, School of Health Sciences, University of Ioannina, University Campus, 5110 Ioannina, Greece
[2] Department of Analytical Chemistry, Faculty of Chemistry, University of Ioannina, 5110 Ioannina, Greece
* Correspondence: vboumba@uoi.gr; Tel.: +30-2651007724 or +30-6974663683

**Abstract:** Brorphine (1-[1-[1-(4-bromophenyl) ethyl]-piperidin-4-yl]-1,3-dihydro-2H-benzo [d]imidazol-2-one) is one of the most recent novel synthetic opioids (NSOs) on the novel psychoactive substances (NPSs) market, involved in over 100 deaths in 2020. Brorphine is a substituted piperidine-benzimidazolone analogue that retains structural similarities to fentanyl, acting as a full agonist at the μ-opioid receptor. Oral Fluid (OF) is an alternative matrix, frequently analyzed for the detection of NPS. Fabric phase sorptive extraction (FPSE) is a superior, green-sample -preparation technology recently applied for drug analysis. This contribution presents the development and validation of a method, based on the application of FPSE and liquid chromatography–tandem mass spectrometry (LC-MS/MS), to determine/quantitate brorphine in OF. The method's linearity ranged between 0.05 and 50 ng/mL ($R^2$ = 0.9993), the bias ranged between 12.0 and 16.8%, and inter- and intra-day precisions ranged between 6.4 and 9.9%. Accuracy and extraction efficiency lied between 65 and 75%. LOD/LOQ were 0.015 ng/mL/0.05 ng/mL. Analyte's post-preparative stability was higher than 95%, while no matrix interferences and carryover between runs were observed. This is the first report introducing the application of FPSE for NPS determination, specifically, the quantification of brorphine in OF, thereby presenting a simple, rapid, sensitive, specific, effective, and reliable procedure engaged to LC-MS/MS that is suitable for routine application and the analysis of more NPSs.

**Keywords:** brorphine; FPSE; LC-MS/MS; NPS; NSO; oral fluid; extraction

## 1. Introduction

Novel synthetic opioids (NSOs) are a subclass of novel psychoactive substances (NPSs) that poses serious risk to the public health in many countries worldwide [1]. NSO class is subdivided into fentanyl- and non-fentanyl- (e.g., U-44700, MT-45) analogs subclasses [2]. The European Monitoring Centre for Drugs and Drug Addiction (EMCDDA) recorded 63 new NSOs and an increased number of NSO-related intoxications (both nonfatal and fatal) in the United States, Europe, Australia, Japan, and Canada between 2009 and 2020, especially in the past few years [2,3].

One of the most recent NSO entries on the NPS market is brorphine, which has been involved in more than 100 deaths in a few months [4]. Brorphine (1-[1-[1-(4-bromophenyl)ethyl]-piperidin-4-yl]-1,3-dihydro-2H-benzo[d]imidazol-2-one) is a substituted, piperidine- benzimidazolone analogue that retains structural similarities (Figure 1) to fentanyl [5]. Brorphine binds to the μ-opioid receptor as an agonist at MOR, being more potent than morphine and less potent than fentanyl [6]. It was firstly synthesized in 1967 by Janssen [7], and its physicochemical properties have not been fully defined yet.

Brorphine was identified on the United States' NPS market in mid-2019 for the first time; later, in June 2020, its identification in Sweden was reported to the EMCDDA [8].

Occasionally, it was found under the street name, "purple heroin" [9]. Moreover, oxycodone tablets adulterated with brorphine, probably purchased via the internet, were implicated in fatal and non-fatal intoxications [10]. The first reported quantification of brorphine in the serum of an intoxicated individual from Belgium was performed by liquid chromatography–high resolution mass spectrometry (LC-HRMS) in February 2020 [11]. Furthermore, a considerable number of intoxications involving brorphine (approximately 20) were reported in mid-2020, and various metabolism studies (in vivo and in vitro) were performed on biological specimens obtained from these cases using LC-MS/MS [4,9]. Currently, brorphine falls under national legislation targeting fentanyl analogues [8,12], having been placed temporarily in Schedule I of the Controlled Substances Act [13,14].

**Figure 1.** Chemical structure of (**A**) brorphine (MW 436.8, pKa 6.61), and (**B**) fentanyl (for comparison).

A wide array of techniques (including GC-MS, liquid chromatography engaged either to DAD or low- and high-resolution mass spectrometers (LC-DAD, LC-MS/MS, LC-HRMS), spectroscopic methods (FT-IR and $^1$H- and $^{13}$C-NM)) has been applied for the determination and identification of brorphine in powder-exhibits [4,8,11]. Also, analytical procedures involving LC-MS/MS have been reported for the quantification of brorphine in conventional biological specimens (blood/serum and urine) [11]. The application of liquid chromatography quadrupole time-of-flight mass spectrometry (LC-QToF-MS) has been used to determine brorphine's metabolites in blood and urine [5]. However, to the best of our knowledge, an up-to-date quantitation of brorphine in oral fluid (OF) has not been reported.

OF is a conventional biological specimen, secreted by salivary glands and composed mainly of water (99.5%), mucus, cells, (such as white blood cells and epithelial cells), molecules, (such as electrolytes, enzymes, and antimicrobial agents), and lysozymes [15]. OF is produced in the salivary glands after "filtering" of blood; therefore, drug levels in it are considered to correspond to free-drug-plasma concentrations [15]. Additionally, OF is slightly more acidic than blood (pH 5.8–6.8), and this results in the ionization of weak basic drugs and higher drug concentrations in OF than in plasma [15]. The advantages of this matrix over conventional biological fluids have established OF as the most suitable alternative matrix to assess recent exposure to psychoactive drugs [15]. Moreover, OF is one of the main biological fluids analyzed for the assessment of different classes of NPS, taking advantage of the progress in extraction and analysis procedures [16–18].

Fabric phase sorptive extraction (FPSE) is a relatively new sample preparation technique that has the potential of wide applications in bioanalysis [19]. The fabric in this technique consists of a fabric substrate, natural or synthetic, that is chemically modified so that an ultra-thin coating, with a hybrid sol–gel and organic–inorganic sorbent, could be formed [20]. The main determinants of the extraction selectivity are the polarity of the polymer, and the hydrophilicity or hydrophobicity of the media [21]. Depending on the physicochemical features of the sol–gel, this microextraction approach can extract analytes with a wide polarity range [22]. FPSE membrane allows direct analytes extraction from the biological substrate, minimizing sample pre-treatment steps. Also, the analytes are desorbed rapidly into organic solvents, from the FPSE media. The aforementioned characteristics of FPSE have demonstrated its leading position as a green-sample-preparation technology of the 21st century. Previous application of FPSE in the analysis of OF in-

cludes the extraction of non-steroidal, anti-inflammatory drugs with the aid of liquid chromatography [23].

The reported implication of brorphine in emergent intoxication cases and the possibility of its widespread use stress the importance of being able to detect this potent NPS in clinical intoxication cases and alternative specimens. The purpose of this contribution is to develop and validate a method based on FPSE and tandem LC-MS/MS, for the identification and quantification of brorphine in OF; the reported procedure has the potential for application of FPSE in the analysis of more NPS in OF.

## 2. Materials and Methods

### 2.1. Chemicals and Reagents

All solvents and reagents used for LC-MS analyses were at least of high-performance liquid chromatography (HPLC) grade. Brorphine HCl ($\geq$98% purity) was obtained from Cayman Chemical Company (Ann Arbor, MI, USA). Ammonium acetate (99%) was purchased from Fluka™ Analytical Standards (Steinheim, Germany). Dichloromethane was purchased from Fisher Scientific (part of ThermoFisher Scientific) (Waltham, MA, USA). Acetonitrile, methanol, water (all UHPLC-MS grade), and formic acid (99%) were obtained from CARLO ERBA Reagents GmbH (Cornaredo, Italy). Ultra-pure water was supplied by an Aquatron Water Still A4000D purification system from Bibby Sterilin Limited (Staffordshire, UK). Whatman microfiber glass filters (110 mm) and Whatman cellulose filter papers (125 mm), which were used for FPSE, were purchased from General Electric (Boston, MA, USA). The organic polymer polyethylene glycol (PEG 300) was from Sigma-Aldrich (Athens, Greece). Trimethoxymethylsilane (MTMS), trifluoroacetic acid (TFA), acetone, sodium hydroxide, and hydrochloric acid were supplied by Merck (Darmstadt, Germany).

### 2.2. Working/Standard Solutions and Calibrators

Stock solution of 1 mg of brorphine dissolved in 1 mL of methanol was stored at $-20\,^{\circ}$C in the dark until analysis. Standard working solutions were prepared daily in methanol. Calibrators at seven concentration levels, 0.05, 0.25, 1.0, 5.0, 10.0, 25.0, 50.0 ng/mL, were prepared by spiking a pool of blank OF. Quality control samples were prepared at 1, 5, and 50 ng/mL in methanol by fortifying blank OF.

### 2.3. Pre-Treatment of Fabric for Sol-Gel Coatings

A Whatman Cellulose circular filter of 125-mm diameter and Whatman Microfiber Glass filter of 110 mm were put in a vial with deionized water and were soaked under constant sonication for 15 min. Then, the procedure was repeated with NaOH (1.0 M, 1 h) and HCl (0.1 M, 1 h). Washing with deionized water was followed in each step. Afterwards, the filter was left to dry at 25 $^{\circ}$C for 24 h, in ambient air, and then followed by sol-gel sorbent coating. In our study, 5 gr of PEG 300 solution was developed, as the sol–gel precursor, using 5 mL of MTMS, 2 mL of TFA catalyst with 5% water, and a mixture of 10 mL of acetone and dichloromethane (50/50 *v/v*). The substrate was immersed into the sol solution for 4 h in room temperature. After sol-gel coating, the fabric was left to dry in the residual sorbents for 24 h; then, it was rinsed with a mixture of acetone and dichloromethane (50/50 *v/v*) under sonication for 30 min, air dried for 30 min, cut into circular pieces with a 1 cm diameter, and stored until use.

### 2.4. Preliminary Experiments

Two types of Whatman filters were examined in the preliminary experiments as an FPSE substrate: microfiber glass filter (FG) and cellulose filter (WC). The extraction times of 10, 20, and 30 min were tested. PH values were investigated at three different levels: acidic (4.5), neutral (7), and alkaline (9). For the most suitable back extraction, solvent methanol, acetonitrile, and their mixture of 50:50 *v/v* were tested. Finally, three different time intervals (5, 10, and 15 min) were examined in order to select the optimal back-extraction time.

### 2.5. Sample Collection and Storage

OF was collected from drug-free donors, by a simple expectoration technique, which allowed it to accumulate in the lower part of the mouth while the subject spat into a pre-weighed test tube every 60 s. Then, the collected OF was vortexed for 30 s and put in an ultrasound bath for 60 s. Finally, aliquots of 10 mL of OF were stored in plastic tubes, frozen until analysis.

### 2.6. FPSE Procedure

One mL of OF was treated with 100 μL of acetonitrile, then repeating vortex for 30 s and ultrasound bath for 60 s, and, finally, centrifugation at 15,000 rpm for 15 min, were applied. The supernatant was put in a tube with a glass microfiber filter (FG) coated with PEG300, and left for 30 min to interact by stirring at 300 rpm, pH = 7. Finally, 150 μL of methanol were added to achieve desorption under gentle stirring at 300 rpm for 10 min.

### 2.7. LC Conditions

Analysis was performed on a Dionex UHPLC system (Thermo Scientific, Waltham, MA, USA) comprised of a degasser, a binary pump, an autosampler, and a column oven. The system was coupled to a Q-Trap 5500™ mass spectrometer (Sciex, Darmstadt, Germany), operated in multiple reaction monitoring (MRM) mode, and equipped with an electrospray ionization (ESI) Turbo V Source operated in positive mode. Separation was performed on an Accupore C18 column (50 mm × 3 mm, 2.6 μm particle size) equipped with a precolumn cartridge (2.1 mm × 0.2 μm) (Thermo Scientific, Waltham, MA, USA), both operated at 30 °C. Mobile phases were: 10 mM of aqueous ammonium acetate adjusted to pH 3.5 with 0.1% formic acid (eluent A) and acetonitrile UHPLC-MS grade with 0.1% formic acid (eluent B), degassed by Elmasonic S ultrasonic, Germany. The autosampler temperature was 5 °C; the injector's needle was rinsed with 200 μL of methanol before and after each injection, and the injected sample volume was 5 μL. Gradient program was as follows: 0.00–1.00 min 12% eluent B; 1.00–2.00 min linear gradient to 50% eluent B, 2.00–3.00 from 50% to 100% eluent B; 3.00–5.00 min remained at 100%; 5.50 min returned to initial conditions; total run time of 6.50 min with a flow rate of 0.500 mL/min.

### 2.8. MS/MS Conditions

The mass spectrometer was equipped with an electrospray ionization source (ESI) operating in positive mode. The applied ESI inlet conditions were as follows: gas 1, nitrogen (55 psi); gas 2, nitrogen (55 psi); ion-spray voltage of 5500 V, positive mode; ion-source temperature, of 550 °C; nitrogen as the curtain gas at 55 psi (Table 1). Optimization of the dwell times and all other settings were performed using the scheduled MRM algorithm incorporated into Sciex Analyst® software version 1.7.1 (Sciex, Darmstadt, Germany), in the automatic quantitative optimization mode. SciexOS 1.6 software 1 (Sciex, Darmstadt, Germany) was applied for data processing.

**Table 1.** MS parameters for the identification and quantitation of brorphine. The ion in bold (218.2) was used for quantification.

| | Q1 Mass (Da) | Q3 Mass (Da) | Dwell Time (ms) | ID | DP (Volts) | EP (Volts) | CE (Volts) | CXP (Volts) |
|---|---|---|---|---|---|---|---|---|
| 1 | 399.9 | **218.2** | 400 | Brorphine 1 | 106 | 10 | 29 | 20 |
| 2 | 399.9 | 182.9 | 400 | Brorphine 2 | 106 | 10 | 33 | 16 |
| 3 | 399.9 | 104.06 | 400 | Brorphine 3 | 106 | 10 | 59 | 12 |

### 2.9. Method Validation

The guidelines expressed by the Scientific Working Group for Forensic Toxicology (SWGTOX) were applied for validation of the analytical procedure [24]. Evaluated parameters were linearity, selectivity, bias, carryover, precision, sensitivity, recovery, matrix effect, and stability.

Selectivity experiments were carried out with blank OF collected from at least ten different drug-free volunteers. The samples were tested to exclude any interference from the matrices and were then pooled. Three blank OF samples were prepared by fortifying with a mixture of 87 drugs (50 ng/mL of each drug) to study selectivity of the method. The evaluation of carryover was performed by analyzing blank matrix samples immediately after running triplicates of high-concentration-control samples (400 ng/mL). The method's linearity was evaluated using triplicates of one of seven concentration levels, 0.05, 0.25, 1.0, 5.0, 10.0, 25.0, and 50.0 ng/mL of brorphine spiked in OF. Sensitivity was expressed by the limit of detection (LOD) and limit of quantification (LOQ), defined as the lowest concentration giving a signal-to-noise ratio of three (S/N = 3) and ten (S/N = 10), respectively. Criteria for identification were a symmetrical peak that eluted at the expected retention time $\pm2\%$, and an ion ratio of quantifier-to-qualifier MRM within $\pm25\%$ of the established standard compound. Both LOD and LOQ were determined by analyzing three fortified OF samples at decreasing concentrations (1–0.001 ng/mL). Intra-day repeatability and inter-day reproducibility (precision), expressed as RSD %, were studied at three concentration levels (1, 5, and 50 ng/mL). Bias was measured in fortified OF samples at three different concentration pools (1, 5, and 50 ng/mL) using a minimum of three separate samples per concentration over five different runs; the maximum value of acceptable bias was $\pm20\%$ at each concentration. Matrix effect (%) was evaluated in low and high concentrations of analyte, of 1 and 50 ng/mL, respectively, and positive results expressed ionization enhancement and negative results ionization suppression. Stability experiments were performed by using fortified OF samples (1 ng/mL and 50 ng/mL) to establish time-zero responses. Then, half vials were refrigerated at $-20\,^\circ$C, and the rest were stored on the autosampler at $5\,^\circ$C. All the vials were then analyzed on LC-MS/MS in triplicate, after two and four weeks, and the average responses were compared to the time-zero ones.

### 2.10. Statistical Analysis

Statistical analysis of the results was carried out using Microsoft EXCEL for Windows to calculate means and standard deviations, and to conduct Student's *t*-test to compare differences between groups (significant at $p < 0.05$).

## 3. Results

### 3.1. Method Development

The applied chromatographic conditions provided a chromatogram of brorphine with a good resolution. A single peak with a retention time of 2.88 min (Figure 2a) and an exact $m/z$ value of 399.9 were obtained; the acquired ion spectrum of brorphine is shown in Figure 2b.

### 3.2. Optimization of the FPSE Procedure for Sample Preparation

The greatest desorption yields were obtained when glass microfiber filter (FG), coated with PEG300, was used; therefore, this combination was chosen for further optimization (Figure 3a). PEG 300 is widely used and was chosen due to its water solubility and hydrophilic properties. The optimum extraction time of 30 min was found to be the most effective (Figure 3b). The extraction time was linked directly to distribution parameters that determined the interaction between the FPSE material and brorphine, indicating a rapid distribution in our case. Neutral pH was shown to have a milder effect on the sample dilution and, consequently, the concentration variation of the analyte (Figure 3c). Methanol was far more efficient than acetonitrile and their mixture (Figure 3d), and the back-extraction-time of 10 min was selected (Figure 3e). Generally, the differences obtained in parameters when applying the different conditions were not statistically significant ($p > 0.05$), and the best performing parameter was chosen to be applied.

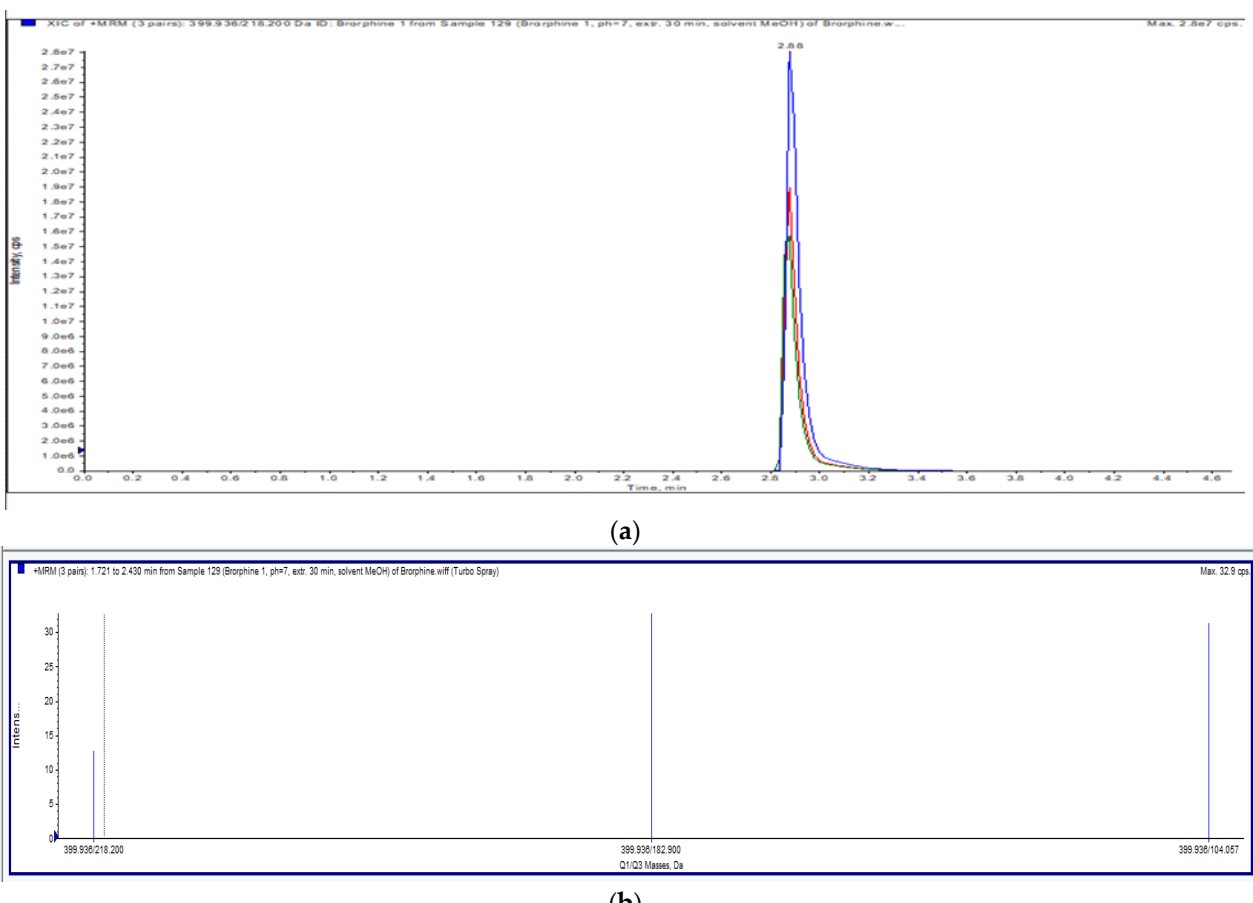

**Figure 2.** (**a**) Total ion chromatogram (Full scan) of brorphine. (**b**) Ion spectra of brorphine.

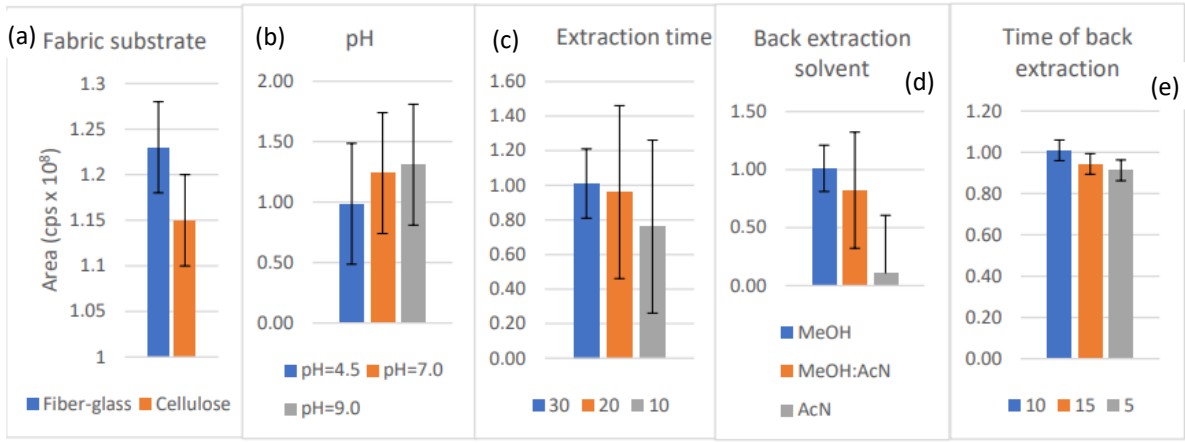

**Figure 3.** (**a**) Influence of coating material on brorphine's yield; (**b**) Influence of pH on brorphine's yield; (**c**) Influence of extraction on brorphine's yield; (**d**) Influence of solvent on brorphine's yield; (**e**) Influence of back-extraction time on brorphine's yield.

### 3.3. Method Validation

The developed method displayed linearity at the specified range between LOQ = 0.05 and 50 ng/mL ($R^2$ = 0.9993). The bias of analyte ranged between 12.0 and 16.8%. Relative standard deviations (RSDs %) for intra-day and inter-day precision were found between 6.4 and 9.9% (Table 2). Accuracy (expressed as recovery %) was in the range of 65–75%. Brorphine's limit of detection (LOD) and limit of quantification (LOQ) were 0.015 ng/mL and 0.05 ng/mL, respectively. Post-preparative stability of the analyte, refrigerated at

−20 °C after two weeks, was 98% and, after four weeks, was 97%. When the analyte was stored at 5 °C after two weeks, its post-preparative stability was 97% and, after four weeks, was 95%. In the selectivity experiments, no matrix interferences and no carryover between analyses during the LC-MS/MS runs were observed.

**Table 2.** Validation parameters of the method for the identification and quantitation of brorphine in OF.

| Analyte | LOD (ng/mL) | LOQ (ng/mL) | Matrix Effect | | Extraction Efficiency | | Intra-Day Precision | | Inter-Day Precision | |
|---|---|---|---|---|---|---|---|---|---|---|
| | | | Low (%) | High (%) | Low (%) | High (%) | Low (%) | High (%) | Low (%) | High (%) |
| Brorphine | 0.015 | 0.05 | −20 | −23 | 65 | 75 | 6.4 | 7.6 | 9.9 | 8.1 |

## 4. Discussion

Although we are not aware of any intoxication case or police case involving brorphine in our country yet, the fact that this substance was identified in some European countries [8–11] indicates that it is possible for it to be spread widely to other countries. Therefore, the development of an analytical methodology to identify and quantify this NSO in clinical cases is essential. Moreover, the application of the FPSE sample preparation technique is employed for the first time to the analysis of an NPS, offering a new tool that could advance forensic research.

The current contribution presents the development and validation of a simple, rapid, and sensitive method for the determination of brorphine in OF, by employing FPSE for the extraction of an analyte, engaged to LC-MS/MS for identification and quantification.

In a series of preliminary experiments, the extraction procedure was optimized regarding coating material, extraction time, pH, and extraction solvent. Methanol was proved to be an efficient back-extraction solvent at the extraction time of 10 min. Adsorption time affects the distribution of analytes between the solvent and the FPSE medium. The required time to reach extraction equilibrium is synergistically affected and reduced by factors such as the porous network of the sol-gel coating, the highly active surface of the FPSE apparatus, and the permeable fabric substrate.

According to literature, brorphine acts as a weak acid [24], so it is expected that the ionized form pKa + 2 (8.61) would provide the ideal outcome. However, it was observed that results with pH = 7 and pH = 9 were comparable to each other. The low viscosity of the solvents allows them to easily penetrate the thin layer of the adsorbent, and decompose quickly and quantitatively brorphine. Our results revealed that a shorter elution time did not demonstrate decent desorption capacity, while increasing the elution time slightly decreased the efficiency, which may be due to the reabsorption of the substances by the FPSE medium.

The developed method shows acceptable analytical characteristics, according to the SWGTOX guidelines [25]. The linearity range in the OF is expected to be related to the concentrations of plasma analyte, a consideration that is confirmed by previous reported results [11]. The external-standard-absolute calibration with certified-reference material is used for the validation experiments of this relatively simple extraction technique [26]. The obtained recoveries, although not excellent (>95%), are considered relatively good and acceptable; they are comparable to relevant results from other studies on brorphine [5,8,11], or, on NPS in general [27]. Matrix effect is acceptable, and is recorded as ion suppression for both low and high concentrations. It is worth mentioning that the limits of detection (LOD) and quantification (LOQ) for brorphine for OF reported in this study are lower than those reported previously for serum and urine (0.05/0.03 ng/mL as the LODs and 0.1 ng/mL as the LOQs for serum and urine, respectively) [5]. The short- and long-term stability of brorphine in OF is acceptable at both −20 °C and at 5 °C.

This is the first report of an analytical procedure for the determination of brorphine in OF; the reported analytical procedure introduces the application of an innovative micro-

extraction technique of FPSE engaged to LC-MS/MS for quantification of an NPS. The characteristics of the FPSE technique show potential for application in the analysis of a wider scale of NPS classes. This relatively simple methodology is effective and reliable for identification and quantitation of brorphine in OF in routine clinical casework.

## 5. Conclusions

The presented method is validated according to international guidelines to identify and quantify brorphine in OF. The application of FPSE for sample preparation engaged to LC-MS/MS resulted in rapid and easy sample preparation, and high sensitivity and selectivity of analysis.

**Author Contributions:** Conceptualization, V.S. and V.B.; methodology, D.F., T.V. and V.B.; validation, D.F., T.V., V.S. and V.B.; investigation, D.F.; data curation, D.F.; writing—original draft preparation, D.F.; writing—review and editing, V.B.; supervision, V.S. and V.B.; project administration, V.B. All authors have read and agreed to the published version of the manuscript.

**Funding:** This research received no external funding.

**Institutional Review Board Statement:** The study was conducted in accordance with the Declaration of Helsinki, and approved by the Institutional Review Board of University of Ioannina/University Hospital of Ioannina (834/18-10-2022).

**Informed Consent Statement:** Informed consent was obtained from all subjects involved in the study.

**Conflicts of Interest:** The authors declare no conflict of interest.

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
