# Peer review of "Application of Fabric Phase Sorptive Extraction (FPSE) Engaged to Tandem LC-MS/MS for Quantification of Brorphine in Oral Fluid"

_jox, doi:10.3390/jox12040025_

Round 1
Reviewer 1 Report
The manuscript describes the development and validation of an analytical method, based on the application of FPSE and tandem LC-MS/MS, to determine brorphine in oral fluid. The publication has mediocre novelty but can be published in the Journal. The authors come from Department of Analytical Chemistry and their validation is perfect. Therefore, my suggestion is acceptance of the manuscript after the following major revisions:
Abstract: Introduce abbreviation NSO and NPS
Section B: LC-HR-MS/MS (LC-QToF-MS) are all abbreviations that must be introcued.
Section B: The abbreviation OF has already been introduced in the text earlier. I suggest that the authors have a thorough read of the manuscript for abbreviations.
Figure 1: The molecule is too big. Please reduce the molecules and add some color to the atoms (if possible), so that it becomes more attractive.
Figure 2 is not readable and must be improved. I suggest to copy paste from the software and use powerpoint to improve the quality. It is a tedious procedure with guaranteed results.
Figure 3 must be improved. I would eliminate the boarders of the figures. Alternatively, fully align them so that they look as one figure and not as 5 separate figures.
Author Response
The authors thank the reviewers for their valuable comments and have decided to undertake revision of the MS according to their suggestions
Reviewer 1
The manuscript describes the development and validation of an analytical method, based on the application of FPSE and tandem LC-MS/MS, to determine brorphine in oral fluid. The publication has mediocre novelty but can be published in the Journal. The authors come from Department of Analytical Chemistry and their validation is perfect. Therefore, my suggestion is acceptance of the manuscript after the following major revisions:
Abstract: Introduce abbreviation NSO and NPS
The suggested abbreviations have been introduced in the revised abstract.
Section B: LC-HR-MS/MS (LC-QToF-MS) are all abbreviations that must be introduced.
The suggested abbreviations have been introduced in the revised MS.
Section B: The abbreviation OF has already been introduced in the text earlier. I suggest that the authors have a thorough read of the manuscript for abbreviations.
The authors apologize for the inconsistency. As suggested abbreviations have been checked and corrected across the text in the revised MS.
Figure 1: The molecule is too big. Please reduce the molecules and add some color to the atoms (if possible), so that it becomes more attractive.
The authors apologize for the pure quality of their figure. As suggested the molecules have been reduced in size and rearranged in the revised figure 1.
Figure 2 is not readable and must be improved. I suggest to copy paste from the software and use powerpoint to improve the quality. It is a tedious procedure with guaranteed results.
The authors thank the reviewer for this comment and suggestion. They followed his/her suggestion, and the figure has been considerably improved in the revised MS.
Figure 3 must be improved. I would eliminate the boarders of the figures. Alternatively, fully align them so that they look as one figure and not as 5 separate figures.
The authors apologize for the pure quality of their figure. As suggested, the graphs have been aligned on one figure in the revised figure 3 that has significantly shown improvement.
Reviewer 2 Report
In my opinion this paper present a very novel technique involving FPSE, a new sample preparation very used in bioanalysis, and LC-MS/MS, to quantify Brorphine in oral fluid. Authos introduce clearly the importance of this molecule, that was identified for the first time on the US NPS market in 2019. Since this year, the number of intoxications involving brorphine has been increased.
Authors propose the quantification of this molecule in oral fluid, in which matrix has not been reported yet.
In my opinion, this paper has a high scientific quality. However, I have some questions that authors must answer and clarify through the paper.
1. Authors comment that the recovery of the developed method is only 65-75%. This percentage is not too high, and In my opinion authors should discuss if there is enough to this method or how would be possible to improve this percentage. A discussion of this aspect should be included in the paper.
2. Do you use any Internal Standard to quantify brorphine? In figure 3, absolute areas are present, and it would be better using relative areas. Why do not use any IS?
3. To carry out the optimization of the methodology, authors optimize different paramets and figure 3 shows in different graphs the results obtained. However, any statistical treatment has been carried out, and it would be better in order to reach concluions. Only applying any statistical treatment, authos could conclude if the differences showed in these graphs are significantly or not. And it is really important to know it.
1.
Author Response
The authors thank the reviewers for their valuable comments and have decided to undertake revision of the MS according to their suggestions.
Reviewer 2
Comments and Suggestions for Authors
In my opinion this paper present a very novel technique involving FPSE, a new sample preparation very used in bioanalysis, and LC-MS/MS, to quantify Brorphine in oral fluid. Authors introduce clearly the importance of this molecule, that was identified for the first time on the US NPS market in 2019. Since this year, the number of intoxications involving brorphine has been increased.
Authors propose the quantification of this molecule in oral fluid, in which matrix has not been reported yet.
In my opinion, this paper has a high scientific quality. However, I have some questions that authors must answer and clarify through the paper.
- Authors comment that the recovery of the developed method is only 65-75%. This percentage is not too high, and in my opinion authors should discuss if there is enough to this method or how would be possible to improve this percentage. A discussion of this aspect should be included in the paper.
The authors thank the reviewer for this substantial comment and agree with him/her that a recovery of 65-75% is not excellent; however, it is considered relatively good and acceptable, as it is reported in relative publications on analysis methods of brorphine or other NPS in biological specimens (e.g. Boumba et al 2017, included as ref [28] in the revised MS).
The following sentence has been rewritten in the text of the revised MS, discussion section, 4th paragraph in the middle: “The obtained recoveries, although not excellent (>95%), are considered relatively good and acceptable, while they are comparable to relevant results from other studies on brorphine [5, 8, and 11] or, on NPS in general [28].”
- Do you use any Internal Standard to quantify brorphine? In figure 3, absolute areas are present, and it would be better using relative areas. Why do not use any IS?
The authors understand the importance of using IS whenever it is possible and support it. However, they respectfully declare that they haven’t use any internal standard for their method, for the following reasons: firstly, because at the time the experiments were performed there wasn’t available a deuterated brorphine (that would have been the ideal IS), and secondly, and more important, the developed methodology using FSPE was considered quite simple; therefore the method could be validated without the use of an IS, but with the external standard absolute calibration with certified reference material, as relative studies suggest (the ref [27] was added in the revised MS). The following sentence has been added in the text of the revised MS, discussion section, 4th paragraph in the middle: “The external standard absolute calibration with certified reference material was used for the validation experiments of this relatively simple extraction technique [27].”
- To carry out the optimization of the methodology, authors optimize different parameters and figure 3 shows in different graphs the results obtained. However, any statistical treatment has been carried out, and it would be better in order to reach conclusions. Only applying any statistical treatment, authors could conclude if the differences showed in these graphs are significantly or not. And it is really important to know it.
The authors thank the reviewer for this comment and made relative changes in the revised MS as suggested.
Specifically, they added the sub-section “2.11 statistical analysis, under the M&M section, in the revised MS: “ 2.11. Statistical analysis
Statistical analysis of the results was carried out using Microsoft EXCEL for windows to calculate means, standard deviations and to conduct Student’s t-test to compare differences between groups (significant at P < 0.05).”
Moreover, the following definition was added in the results section, 3.2 sub-section, last sentence: “Generally, the differences obtained in parameters when applying the different conditions were not statistically significant (p>0.05) and the best performing parameter was chosen to apply.”
Round 2
Reviewer 1 Report
I checked the changes that the authors introduced in the manuscript. I believe that the manuscript must be accepted for publication in its current form.
Author Response
The authors thank the reviewer for is/her suggestion.